# The Development of Herbicide Resistance Crop Plants Using CRISPR/Cas9-Mediated Gene Editing

**DOI:** 10.3390/genes12060912

**Published:** 2021-06-12

**Authors:** Huirong Dong, Yong Huang, Kejian Wang

**Affiliations:** State Key Laboratory of Rice Biology, China National Rice Research Institute, Chinese Academy of Agricultural Sciences, Hangzhou 310006, China; donghuirong1@126.com (H.D.); huangyong@caas.cn (Y.H.)

**Keywords:** CRISPR/Cas, genome editing, herbicide resistance, application

## Abstract

The rapid increase in herbicide-resistant weeds creates a huge challenge to global food security because it can reduce crop production, causing considerable losses. Combined with a lack of novel herbicides, cultivating herbicide-resistant crops becomes an effective strategy to control weeds because of reduced crop phytotoxicity, and it expands the herbicidal spectrum. Recently developed clustered regularly interspaced short palindromic repeat/CRISPR-associated protein (CRISPR/Cas)-mediated genome editing techniques enable efficiently targeted modification and hold great potential in creating desired plants with herbicide resistance. In the present review, we briefly summarize the mechanism responsible for herbicide resistance in plants and then discuss the applications of traditional mutagenesis and transgenic breeding in cultivating herbicide-resistant crops. We mainly emphasize the development and use of CRISPR/Cas technology in herbicide-resistant crop improvement. Finally, we discuss the future applications of the CRISPR/Cas system for developing herbicide-resistant crops.

## 1. Introduction

The world population is expected to reach 10 billion by the year 2050, an increase of 34%. As a result, the global grain yield needs to increase by 70%, according to the Food and Agriculture Organization of the United Nations, to meet the demand of the world population [1,2]. However, global food security has suffered other challenges because of climate change, reduced arable land, scarcity of water, and biotic and abiotic stresses [3], which could affect crop production and cause enormous losses.

Weed damage is one of the main obstacles in crop production [4]. The presence of weeds in farmland will compete with crops for growth space, water, fertilizer, sunlight, and spread pests and diseases directly or indirectly [5,6], thereby inhibiting crop growth, reducing crop yields, and even seriously affecting the quality of crops [7,8]. Additionally, several weed seeds or pollen contain toxins; adulteration in crop seeds can cause poisoning to humans and animals. At present, chemical herbicides have been widely used in agronomic crops as the primary method to control weeds due to their economic and effective effects [9]. However, herbicide-resistant weeds quickly followed due to the extensive and recurrent use of the same herbicides. Cultivation of herbicide-resistant crops is an effective measure to control weeds, which can reduce crop phytotoxicity due to herbicide application [10], expand the herbicidal spectrum [11], and reduce the cost of weeding [12]. Thus, developing herbicide resistance crops is the most efficient strategy to control weed proliferation and tremendously increase crop productivity [13].

Previously, traditional mutagenesis breeding played an important role in improving herbicide-resistant crops, but it is labor-intensive and time-consuming. After the transgenic techniques became available, it has been successfully used for crop improvement in the past few decades [14,15]. Moreover, transgenic herbicide-resistant crops have also increased dramatically [16]. However, due to the transfer of foreign genes, the promotion and use of genetically modified products are restricted, and the wide application of this approach is also limited [17]. Therefore, a more precise breeding technique is highly required.

Since the clustered regularly interspaced short palindromic repeats/CRISPR-associated (CRISPR/Cas) genetic scissors were discovered in 2012 [18,19], CRISPR/Cas-based techniques have been successfully used to accelerate plant breeding for desirable traits. These traits include increased yield and nutritional value, stress tolerance, and pest and herbicide resistance because of its simplicity, flexibility, and high specificity. The CRISPR/Cas system generates double-strand breaks (DSBs) at target loci [19], and two main pathways can repair the DSBs, namely, nonhomologous end-joining (NHEJ), an error-prone repair system that often allows the introduction of deletions, insertions, or substitutions [20], and homology-directed repair (HDR) when a donor template with the homologous sequence is available [21]. NHEJ is the preferred repair pathway, because it does not require a homologous repair template [22]. If the target sequence is near microhomologous sequences, then the DSBs may be repaired through microhomology-mediated end-joining, resulting in fragment deletions of the plant genome with higher efficiency [23,24].

In this review, we briefly summarize the mechanism and site mutations that confer resistance to herbicides in plants and discuss the differences between traditional mutagenesis and transgenic breeding in cultivating herbicide-resistant crops. Importantly, we focus on the development and use of CRISPR/Cas technology in herbicide-resistant crop improvement. Additionally, we discuss the potential applications of the methods for herbicide-resistant crops. We hope that this review provides new insights for the future directions of genome editing technology to improve crop genetic breeding.

## 2. Site Mutations Confer Resistance to Herbicides in Plants

Herbicides can interfere with and inhibit the metabolic processes of plants, thus causing plant death. Therefore, they are widely used for weed control in crop fields. However, the prolonged and extensive use of certain herbicides or groups of herbicides has consequently led to the evolution of resistance in many weed species, and the resistance to herbicides in weeds is increasing rapidly worldwide. Tolerance to acetolactate synthase (*ALS*)-inhibiting herbicides, acetyl-CoA carboxylase (*ACCase*)-inhibiting herbicides and 5-enolpyruvylshikimate-3-phosphate (*EPSPS*) synthase-inhibiting herbicides have been reported more frequently, especially ALS-inhibiting herbicides, which account for one-third of all weed resistance [25].

Generally, herbicide resistance can be achieved through several mechanisms, such as target site mutation, target site gene amplification, increased herbicide detoxification, and metabolism [26]. Target site mutation is the primary resistance mechanism and has been successfully applied to creating herbicide-resistant germplasm in many crop species. For the development of herbicide-resistant plants, it is very important to select target genes associated with important resistance traits. Notably, the *ALS* gene, *ACCase* gene, and *EPSPS* gene have provided a wealth of potential genes to produce herbicide-resistant crops.

ALS (EC 4.1.3.18) is the key enzyme that catalyzes the first step in the biosynthesis of the three branched-chain amino acids (leucine, isoleucine, and valine) [27]. This is the target enzyme of ALS-inhibiting herbicides with dissimilar chemistries, including sulfonylureas (SU), imidazolinones (IMI), pyrimidinylthiobenzoates (PTB), sulfonylaminocarbonyl-triazoli-nones (SCT), and triazolopyrimidines (TP) [28]. So far, eight variant *ALS* genes have been discovered in weeds as natural mutations conferring tolerance to ALS-inhibiting herbicides involve the amino acids residues Ala122, Pro197, Ala205, Asp376, Arg377, Trp574, Ser653, and Gly654, in which the amino acid number is based on the *Arabidopsis ALS* gene [25].

ACCase (EC 6.4.1.2) is a crucial enzyme for fatty acid biosynthesis pathways. ACCase-inhibiting herbicides such as aryloxyphenoxypropionate (APP), cyclohexanedione (CHD), and phenylpyrazoline (PPZ) inhibit fatty acid synthesis in plants leading to plant death [29]. Many weeds acquire resistance to ACCase inhibitors due to amino acid mutations at seven positions: Ile1781, Trp1999, Trp2027, Ile2041, Asp2078, Cys2088, and Gly2096 [30].

Glyphosate is a broad-spectrum, non-selective herbicide that has become the globally dominant herbicide [31] that inhibits EPSPS (EC 2.5.1.19), a critical enzyme involved in the biosynthesis of aromatic amino acids [32]. As a result of intense glyphosate use, resistance to this herbicide has been confirmed in many weed species. Target site mutations occurring at Thr102, Ala103, and Pro106 of the *EPSPS* gene are known to endow glyphosate resistance in several fields that evolved resistant weed species [33].

## 3. Transgenic and Mutagenesis Breeding of Herbicide-Resistant Crops

Since weeds are a major threat to global food production, cultivating herbicide resistance crops has become an integrated part of modern weed management, which can decrease the phytotoxicity of herbicides to crops, improve the efficiency of the chemical weeding, and reduce the cost of weeding [34]. Over the last few decades, transgenic and traditional mutagenesis breeding (Figure 1A,B,D), have demonstrated their power in generating commercially available herbicide-resistant crops [35].

Transgenic breeding technology should introduce exogenous elite genes found in other species and organisms. Currently, most commercialized herbicide-resistant crops were obtained through this technology and account for approximately 25% of all genetically modified (GM) crops [36]. Recent advances indicate that most of the increased yield in transgenic herbicide-resistant crops is attributed to glyphosate-resistant soybeans, maize, rice, wheat, canola, cotton, and sugar beets [11]. The *CP4* gene, glyphosate N-acetyltransferase (*GAT*) gene, and glyphosate oxidoreductase (*GOX*) gene were the primary genes that contributed to glyphosate resistance in crops [37,38]. Transgenic glufosinate-resistant crops with the phosphinothricin N-acetyltransferase (*PAT*) gene or bialaphos resistance (*Bar*) gene have also been illegally commercialized [39,40,41]. Crops resistant to other herbicide types, such as ALS-inhibiting herbicides, ACCase-inhibiting herbicides, synthetic auxin herbicides, and HPPD-inhibiting herbicides, can also be developed by this method. However, the development of GM crops is subject to several factors: cumbersome operation, labor-intensive, high cost, and regulatory restriction [42,43].

In contrast to the transgenic approach, herbicide-resistant crops generated by traditional mutagenesis breeding are considered non-GM. From 1992 to the present, the development and promotion of herbicide-resistant crops through mutagenesis have made rapid progress. A series of ALS inhibitor-resistant maize, wheat, rice, oilseed rape, and sunflower were developed by mutagenesis breeding; these crops are commercialized as Clearfield^®^ crops [44,45]. Notably, the first commercialized ACCase-inhibiting herbicide resistance crop was a sethoxydim-resistant corn, with an altered ACCase activity created through tissue culture [46]. The non-GM ACCase-inhibiting herbicide resistance rice named Provisia™ was also promoted globally [47]. Both the Clearfield^®^ series and Provisia series are artificially induced. Although the non-GM crops to be introduced are commercially available, the application of mutagenesis breeding has remained restricted due to its low mutation frequency and random creation of mutations, making it nearly impossible to induce multiple specific mutations for acquiring herbicide resistance simultaneously [36].

## 4. Herbicide-Resistant Crops Generated by CRISPR/Cas System

With the advent of modern molecular biology, more recently developed CRISPR/Cas-mediated genome editing techniques offer an effective alternative method for inducing genetic modification in various crops [48], exhibiting great potential in accelerating the development of improved crop varieties (Table 1). Thus, genome editing technology could improve precise modifications of DNA sequences that correlated with herbicide resistance (Figure 1C,D).

### 4.1. NHEJ Pathway to Improve Herbicide Resistance in Crops

Despite the dominance of NHEJ pathway during the DSBs repair, only a handful of editing events about creating herbicide-resistant crops have been reported in which endogenous plant genes were accurately modified this way.

Zhang et al. [83] obtained the paraquat-resistant lettuce with small or large deletion through editing the uORF of *LsGGP2*, a gene link to the production of ascorbic acid that increases oxidative stress tolerance in plant cells. This indicated that the gene editing introduced by the NHEJ repair pathway has measurable effects on the creation of herbicide-resistant crops. Except for the multiple base deletions, small insertions or deletions (indels), such as +1/−1 bp indels, often occurred in the period of the NHEJ repair pathway. Fortunately, resistance to ALS-inhibiting herbicide and ACCase-inhibiting herbicide were improved in *Arabidopsis* via the deletion of a base followed by the insertion of a different one [84], and the NHEJ repair strategy for the improvement of herbicide-resistant germplasm may also be applicable to crops.

Efficient intron-mediated site-specific gene replacement and insertion can also be generated through the NHEJ pathway using the CRISPR/Cas9 system. Depending on double amino acid substitution, T102I and P106S (TIPS) in the conserved motif of the endogenous *EPSPS* gene lead to resistance to glyphosate in goosegrass [82]. Scientists developed TIPS double amino acid substitutions in rice plants harboring the *OsEPSPS* gene, and the intended substitutions demonstrated resistance to glyphosate using the NHEJ pathway to generate gene replacements and insertions [65].

Herbicide resistance reported in plants indicated that the NHEJ pathway is suitable for developing important traits in crops. Nevertheless, their applications in crop improvements are rather limited, because they often yielded insertions or deletions that largely resulted in loss-of-function mutations [85], and many agriculturally important traits are conferred by point mutations, gene replacements, or gene knock-in by homologous recombination. Thus, the development of genome editing that enables gene replacement rather than gene inactivation will greatly facilitate plant breeding [76].

### 4.2. Development of Herbicide Resistance Crops via HDR Pathway

In contrast to NHEJ, the HDR pathway can modify endogenous genes precisely with targeted gene insertion or gene replacements, whereas HDR events remain much lower than that of NHEJ [86]. The recent application of HDR-dependent genome editing using the CRISPR system can potentially provide a feasible approach in plant breeding.

The resistant herbicide maize and soybeans were obtained through the HDR pathway mediated by the CRISPR/Cas9 technology [87,88]. Additionally, novel rice germplasms with bispyribac-sodium resistance were obtained by introducing point mutations at the 548th and 627th amino acid positions of the rice *ALS* gene [51,52]. Besides, a chimeric Cas9-VirD2 protein that combines Cas9 and VirD2, a Vir protein that cleaves the bottom strands of the Ti plasmid in the left and right border, was developed to enhance HDR efficiency in plants. Furthermore, precise *OsALS* allele modification that yields herbicide-resistant rice was successfully gained by employing this system [66]. Similarly, the glyphosate-resistant trait was optimized in rapeseed by introducing the donor template and a geminiviral replicon into plant cells [67]. Otherwise, the ALS inhibitor-resistant rice was generated with chimeric single-guide RNA (cgRNA) molecules composed of target site-specificity and repair template sequences [68]. The herbicide resistance trait in flax was also discovered owing to the combination of single-stranded oligonucleotides (ssODN) and CRISPR/Cas9 [77].

The development of the CRISPR/Cpf1 system has further expanded the application scope of genome editing technology based on the HDR pathway because of its long 5′-protruding ends [69], which may facilitate the pairing and insertion of repair templates, although the system has the disadvantage that nonspecific cleavage activity of single-stranded DNAs [78]. Scientists performed allelic replacement of the wild-type *ALS* gene with the intended mutations that carries two discrete point mutations, thereby conferring herbicide resistance in rice plants [89]. However, the efficiency is still very low through homology-mediated repair pathways. They improved the CRISPR/Cpf1 system and successfully the high-efficiency editing, simultaneously obtaining herbicide-resistant rice plants [70]. Prime editing is a novel and universal CRISPR/Cas-derived precision genome editing technology without exogenous donor DNA repair templates [71]. Jiang et al. [90] and Butt et al. [72] obtained herbicide resistance maize and rice through this editing system, respectively.

Notwithstanding the above reports, several limitations are still present, including the dominance of the NHEJ repair pathway, the difficulty in delivering sufficient repair templates [60,91], and susceptibility to degradation by cellular nucleases [92].

### 4.3. Improving Herbicide Resistance through Base Editing

Base editing enables irreversible conversion of base-pair without requiring double-stranded DNA breaks or donor repair templates [93]. So far, the base editors to convert C:G > T:A mutations and A:T > G:C base pairs at target loci have been developed and named cytidine-deaminase-mediated base editor (CBE) or adenine-deaminase-mediated base editor (ABE), respectively [93,94]. Base editors make it possible to correct a substantial fraction of herbicide resistance-associated SNPs. The two base editors have been introduced into several genes, including the *ALS* gene, *ACCase* gene, *EPSPS* genes, and other genes related to herbicide resistance in crops.

Fortunately, developing crop varieties harboring herbicide-resistant mutations that render crop tolerance to ALS-inhibiting herbicides by the CBE has been successfully applied in various crop species, including rice [61,95], maize [64], wheat [53,54], watermelon [55], oilseed rape [56], tobacco [57], tomato, and potato [58]. Similarly, crop tolerance to ACCase-inhibiting herbicides was also found in wheat via the CBE editor [53]. To improve the editing efficiency of CRISPR/Cas technology, the target-activation induced cytidine deaminase (Target-AID) was developed for improving multiplex traits that could accelerate crop improvement at one time in combination with targeted base editing. Furthermore, this base editing system has been applied to produce herbicide-resistant germplasm in rice plant [49,59].

Conversely, the ABEs also exhibited powerful potential for developing novel germplasms to confer resistance to herbicides. Li et al. [50] optimized an ABE for application in plant systems with an evolved tRNA adenosine deaminase, achieved targeted conversion of adenine to guanine, and produced a haloxyfop-R-methyl-resistant rice plant [50]. Additionally, the mutation of the *OsTubA2* gene can also be rapidly introduced to confer resistance to both trifluralin and pendimethalin herbicides in rice using CRIPSR-mediated adenine base editors [73]. Recently, Yan et al. [79] simultaneously developed novel SNPs in four endogenous herbicide target genes (*OsALS1*, *OsGS1*, *OsTubA2*, *OsACC*) with induced efficient A-to-G conversion using new TadA variants, named TadA9.

Directed evolution has proved to be an effective strategy for accelerating the improvement of crop traits, because genetic diversity is artificially increased. Combined with a sgRNA library, the CRISPR/Cas9 system can generate considerable gene variants, thus driving the directed evolution of proteins [62]. Butt et al. [96] developed a CRISPR/Cas-based directed evolution system in rice, and the spliceosome component SF3B1 mutant variants conferred variable levels of resistance to herboxidiene (GEX1A) [80,96,97]. Moreover, a base editing-mediated gene evolution method was present with both the CBE editor and ABE editor alongside a sgRNA library to generate various nucleotide changes in target genomic regions. Consequently, four different novel amino acid substitutions that have never been reported formerly were identified in *OsALS*, exhibiting various resistance levels to bispyribac-sodium belonging to the ALS inhibitor [81]. Meanwhile, a dual base editing system that fused cytidine deaminase with adenosine deaminase named STEMEs was developed, enabling C:G > T:A and A:T > G:C substitutions in the same target sequence, provided directed evolution of endogenous genes by improved saturated mutagenesis. The saturated mutagenesis of the *ACCase* gene in rice plants by this dual base editing system can generate mutations associated with ACCase inhibitor, haloxyfop [63]. Accordingly, to exploit dominant mutations endowing resistance to ACCase-inhibiting herbicide, 141 sgRNAs were designed in the carboxyltransferase domain of the *ACCase* gene. As a result, a novel W2125S mutation was produced by CRISPR-mediated directed evolution to confer APP herbicide resistance in rice [74].

## 5. Conclusions and Prospect

The CRISPR/Cas technology has an enormous potential as a precise and straightforward genome editing tool to edit the plant genome, which is required for herbicide resistance crop improvement [75]. However, there have been many reports of successfully using this technology to create herbicide-resistant crop germplasm. Unfortunately, several challenges remain to be addressed, and further efforts are required to overcome these.

One of the primary challenges is whether the CRISPR/Cas system could obtain other potential genes for creating herbicide-resistant crops. For example, the Thr 239 Ile and Leu 136 Phe mutations of α-tubulin could endow resistance to dinitroaniline, and both mutations were associated with cross-resistance to benzoic acids, benzamides, and carbamates [98,99,100]. Additionally, the Met 268 Thr mutation association with resistance to dinitroaniline was also demonstrated in *Eleusine indica* and *Setaria viridis*, respectively [101]. Furthermore, this point mutation has been used to develop herbicide resistance in rice [73]. However, studies on the discovery and of these potential genes in herbicide-resistant crops are still lacking.

Secondly, among the creation of herbicide-resistant crop germplasms, only the resistance to ALS-inhibiting herbicides, ACCase-inhibiting herbicides, and glyphosate in crops has achieved greater success. However, for the 4-hydroxyphenyl pyruvate dioxygenase and protoporhyrinogen oxidase inhibiting herbicide, studies on their widespread use and effective control of weeds are deficient.

Finally, although non-selective herbicides, such as glyphosate and glufosinate, have a broad-spectrum of herbicidal properties, only a few studies have reported that they improve crop resistance. Currently, the development of herbicide-resistant crops mainly focuses on selective herbicides. Moreover, creating crops that are resistant to multiple types of herbicide simultaneously requires research. Therefore, it is significant to breed non-selective herbicides or multiple herbicide-resistant crops.

At present, most of the research focuses on the cultivation of herbicide-resistant crop varieties, but the characteristics of herbicide resistance have not been fully combined with other breeding strategies. These strategies include (1) cultivating crops resistant to herbicides and disease and insect pests, and (2) optimizing assisted cross-breeding using the characteristics of herbicide resistance. For example, one strategy would involve spraying herbicides at the seedling stage to improve the purity of hybrid rice. With the continuous improvement of the gene editing technology, the cultivation of herbicide-resistant crop varieties will progress.

## Figures and Tables

**Figure 1 genes-12-00912-f001:**
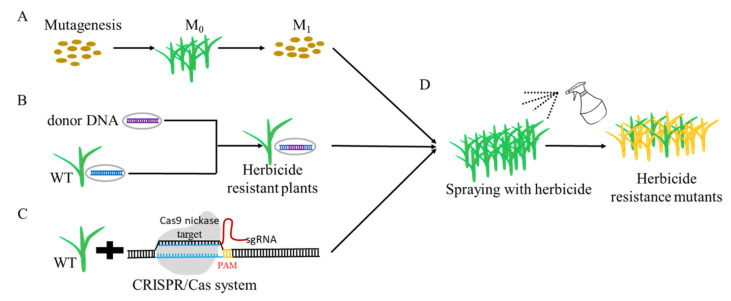
Creation of herbicide-resistant breeding crops. (**A**) Crop breeding by mutagenesis method. (**B**) Crop improvement by transgenic technology. (**C**) Crop genetic breeding through CRISPR/Cas system. (**D**) Screening resistant-plants with herbicide treatment.

**Table 1 genes-12-00912-t001:** The herbicide-resistant plants generated by CRISPR/Cas9-mediated gene editing.

Gene	Mutation Sites	Herbicide Resistance Plants	Repair Pathway	References
ALS	Ala96	rice	CBE	[49,50]
	Pro165	maize	HDR	[51]
	Pro178	soybean	HDR	[52]
	Pro165	maize	CBE	[53]
	Pro174	wheat	CBE	[54,55]
	Pro174	watermelon	CBE	[56]
	Pro197	oilseed rape	CBE	[57]
	Pro194	tobacco	CBE	[58]
	Pro197	tomato and potato	CBE	[59]
	Trp548	rice	HDR	[60]
	Gly628	rice	CBE	[61]
	Ser627	rice	ABE	[62]
	/	rice	CDE	[63]
	Pro171/Gly628	rice	CBE	[64]
	Trp574/Ser653	*Arabidopsis*	NHEJ	[65]
	Trp548/Ser627	rice	HDR	[66,67,68,69,70,71]
	Trp542/Ser621	maize	HDR	[72]
ACCase	Trp2038	*Arabidopsis*	NHEJ	[65]
	Cys2186	rice	ABE	[62,73]
	Ala1992	wheat	CBE	[54]
	/	rice	CDE	[74,75]
EPSPS	Thr102/Pro106	rice	NHEJ-HDR	[76]
	Thr102/Pro106	rapeseed	HDR	[77]
	Thr178/Pro182	flax	HDR	[78]
TubA2	Met268	rice	ABE	[79]
	Met268	rice	ABE	[62]
SF3B1	/	rice	CDE	[80,81]
LsGGP	uORF	lettuce	NHEJ	[82]

Abbreviations: NHEJ-HDR means targeting gene replacement strategy via the NHEJ pathway using CRISPR–Cas9; CBE means cytosine base editor; ABE means adenine base editor; CDE means CRISPR/Cas-mediated direction evolution.

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
