# Peer review of "The Development of Herbicide Resistance Crop Plants Using CRISPR/Cas9-Mediated Gene Editing"

_genes, 2021, doi:10.3390/genes12060912_

Round 1
Reviewer 1 Report
The authors has choosen a good topic for writing a review which scientifically relevant and will go along way towards further research and development for herbicide tolerance in crop plant.
But I have major reservation on how the review has been written or arranged.
First of all the 2nd subheading 'Site mutations confers resistance to herbicide in plants'. In this section the authors have described the genes in which mutations has been observed naturally and the site of mutations. But they have not discussed in which plants these mutations were observed were these unwanted weeds or crop plants. If it is in unwanted weeds what is the remedy to get rid of such plants? How this knowledge of natural mutations have been applied in crop plants to make them herbicide resistant?
Secondly, in the sub-heading 'Traditional Breeding of herbicide-resistant crops', as evident from the heading this parah should deal with traditional breeding. Instead it is full with transgenic breeding and generation of transgenic using various genes.
Thirdly, in figure 1, I can find only (D) but no (A), (B), (C) whereas they are mentioned in manuscript several times.
Fourthly, in table 1, two additional types of mutations which has been reported for EPSPS is missed out (references for those are present underneath). This might be the case for other genes also please conduct a review of literature again so that all the cases are taken into account.
García, M. J., Palma-Bautista, C., Rojano-Delgado, A. M., Bracamonte, E., Portugal, J., Alcántara-de la Cruz, R., & De Prado, R. (2019). The triple amino acid substitution TAP-IVS in the EPSPS gene confers high glyphosate resistance to the superweed Amaranthus hybridus. International journal of molecular sciences, 20(10), 2396.
Eichholtz, D. A., Gasser, C. S., & Kishore, G. M. (2001). U.S. Patent No. 6,225,114. Washington, DC: U.S. Patent and Trademark Office.
Author Response
- First of all the 2nd subheading 'Site mutations confers resistance to herbicide in plants'. In this section the authors have described the genes in which mutations has been observed naturally and the site of mutations. But they have not discussed in which plants these mutations were observed were these unwanted weeds or crop plants. If it is in unwanted weeds what is the remedy to get rid of such plants? How this knowledge of natural mutations have been applied in crop plants to make them herbicide resistant?
Response: Thanks for your suggestions. The site of mutations in the section “Site mutations confers resistance to herbicide in plants” were observed in weeds (See the line 90, 97 and 106). To develop the herbicide resistant crops, we need to know the amino acid change that could endow resistance to herbicides. But these mutations have been reported in a variety of weeds, and in this article, we focus more on the application of the reported mutations that make insensitive to the corresponding herbicides, so we did not list the unwanted weeds in detailed. Since weed is one of the major threats to reduce production, developing crop varieties with herbicide resistance may be cost effective tools for helping farmers to manage weeds, the detailed description can be found in line 37-41. In order to use the natural mutations found in weeds for herbicide resistant crops, we need to search the homologous genes in crops that produce herbicide resistance with genetic mutations in weeds. Then we use the CRISPR/Cas9 technology to produce double strand breaks or make base pairs conversion at the amino acid site that could show resistance to herbicide with mutation, which can endow the gene certain herbicide resistance, so the herbicide resistant crops can be developed. The detailed methods have been explained in the “Herbicide‑Resistant Crops Generated by CRISPR/Cas System”.
- Secondly, in the sub-heading 'Traditional Breeding of herbicide-resistant crops', as evident from the heading this parah should deal with traditional breeding. Instead, it is full with transgenic breeding and generation of transgenic using various genes.
Response: Thanks for your suggestions. We have modified the sub-heading to make it consistent with the content of this part in the revised manuscript (See the Line 107).
- Thirdly, in figure 1, I can find only (D) but no (A), (B), (C) whereas they are mentioned in manuscript several times.
Response: Thank you very much for nice comments. We have marked the A, B and C in the figure in the revised manuscript.
- Fourthly, in table 1, two additional types of mutations which has been reported for EPSPS is missed out (references for those are present underneath). This might be the case for other genes also please conduct a review of literature again so that all the cases are taken into account.
Response: Thank you for pointing this out. The plants showed in table 1 was the herbicide-resistant crops generated by CRISPR/Cas9-mediated gene editing, the two additional types of mutations were found in weeds, so we did not list in this table.

Reviewer 2 Report
In the manuscript “The Development of Herbicide Resistance Crop Plants using CRISPR/Cas9‑Mediated Gene Editing” by Huirong Dong et al., authors have precisely described the role of CRISPR Cas9 in generating herbicide resistant crops which is needed with the increasing world population. In the beginning authors have mentioned about the traditional methods used for mutagenesis or crop improvement, to give the background information. The mechanism of CRISPR Cas9 has also been described briefly. The authors have clearly discussed the challenges and future perspective. The manuscript is very well written, with smooth transition between different sections.
I suggest a few minor corrections below:
Line 87: Please replace the comma with a full stop after [27].
Figure 1: Please mark A, B and C.
Line 140-141: Please add ‘are’ in the sentence- Although the non-GM crops to be introduced are commercially available.
Line 214-215: Please rephrase the sentence.
While mentioning Cpf1, please also mention about its nonspecific cleavage activity against ssDNA which can impact insertion of the repair template.
Line 227: Please replace the comma with a full stop after [73, 74].
Author Response
- Line 87: Please replace the comma with a full stop after [27].
Response: Thanks for your comments. We have corrected it in the revised manuscript (See the Line 87).
- Figure 1: Please mark A, B and C.
Response: Thanks. We have added the mark in the revised manuscript
3) Line 140-141: Please add ‘are’ in the sentence- Although the non-GM crops to be introduced are commercially available.
Response: Thanks for your pointing. We have added it in the revised manuscript (See the Line 141).
4)Line 214-215: Please rephrase the sentence. While mentioning Cpf1, please also mention about its nonspecific cleavage activity against ssDNA which can impact insertion of the repair template.
Response: Thank you for pointing this out. We have corrected the description in the revised manuscript (See the Lines 211-213).
- Line 227: Please replace the comma with a full stop after [73, 74].
Response: Thanks for your comments. We have corrected it in the revised manuscript (See the Line 230).

Round 2
Reviewer 1 Report
I am satisfied with the author's responses
Author Response
Thank you very much!